# Gut–Liver Axis and Inflammasome Activation in Cholangiocyte Pathophysiology

**DOI:** 10.3390/cells9030736

**Published:** 2020-03-17

**Authors:** Luca Maroni, Elisabetta Ninfole, Claudio Pinto, Antonio Benedetti, Marco Marzioni

**Affiliations:** Department of Gastroenterology and Hepatology, Università Politecnica delle Marche, 60126 Ancona, Italy; elisabettaninfole@gmail.com (E.N.); pintoclaudio86@gmail.com (C.P.); a.benedetti@univpm.it (A.B.); m.marzioni@univpm.it (M.M.)

**Keywords:** inflammasome, NLRP3, cholangiocyte, gut–liver axis

## Abstract

The Nlrp3 inflammasome is a multiprotein complex activated by a number of bacterial products or danger signals and is involved in the regulation of inflammatory processes through caspase-1 activation. The Nlrp3 is expressed in immune cells but also in hepatocytes and cholangiocytes, where it appears to be involved in regulation of biliary damage, epithelial barrier integrity and development of fibrosis. Activation of the pathways of innate immunity is crucial in the pathophysiology of hepatobiliary diseases, given the strong link between the gut and the liver. The liver secretes bile acids, which influence the bacterial composition of the gut microbiota and, in turn, are heavily modified by microbial metabolism. Alterations of this balance, as for the development of dysbiosis, may deeply influence the composition of the bacterial products that reach the liver and are able to activate a number of intracellular pathways. This alteration may be particularly important in the pathogenesis of cholangiopathies and, in particular, of primary sclerosing cholangitis, given its strong association with inflammatory bowel disease. In the present review, we summarize current knowledge on the gut–liver axis in cholangiopathies and discuss the role of Nlrp3 inflammasome activation in cholestatic conditions.

## 1. Introduction

The hepatobiliary system is constantly exposed to a wide variety of antigens. A number of environmental toxins ingested with the diet, food antigens and also bacterial products known as pathogen-associated molecular patterns (PAMPs) are, indeed, absorbed from the gut mucosa and reach the liver parenchyma through the portal circulation. The liver possesses unique features that maintain an immune tolerance against antigens, which is mainly based on its strong innate immune system [1]. While such mechanisms are important in maintaining immunological functions, in pathologic conditions the activation of a number of innate immunity receptors may contribute to disease development and progression. Intestinal dysbiosis, which is often associated to a number of liver diseases, may be particularly relevant in this setting [2].

Cholangiocytes are the epithelial cells that line the ducts and are the target of a group of diseases named cholangiopathies, such as primary sclerosing cholangitis (PSC) and primary biliary cholangitis (PBC) [3]. Despite the fact that bile is thought to be sterile in normal conditions, PAMPs, such as lipopolysaccharide (LPS), lipoteichoic acid and bacterial DNA fragments, may be detected in human bile [4,5,6]. Not surprisingly, it is well known that cholangiocytes express a wide array of receptors of innate immunity [7]. Human biliary cells have been shown to express the mRNAs for all ten human Toll-like receptors (TLRs) [8]. Immunohistochemical staining confirmed that TLRs are present in the intrahepatic biliary tree of normal and diseased human livers [9]. Moreover, LPS stimulation in cholangiocytes is able to activate TLR4 and cause the secretion of Il-6 and Il-8 via NF-κB and MAPK signaling pathways, supporting a functional role of TLRs in biliary pathophysiology [10]. Such activation may be particularly important in the setting of liver cirrhosis, in which portal levels of LPS have been shown to be increased compared to venous blood, with even higher levels in case of acute decompensation of liver function [11,12]. 

Inflammasomes are cytosolic multiprotein complexes of the innate immune system responsible for the activation of inflammatory responses [13]. In the cell cytoplasm, the oligomerization of multiple inflammasome components causes caspase-1 activation, which subsequently triggers the maturation of the pro-inflammatory cytokines IL-1β and IL-18, and the initiation of the inflammatory cell death termed pyroptosis [14]. Inflammasomes have mainly been studied in the professional immune cells of the innate immune system, such as macrophages. Recent studies have shown the role of NRLP3 inflammasome in severe liver inflammation, hepatocyte pyroptosis and hepatic stellate cell (HSC) activation with collagen deposition in mice [15]. The NLRP3 inflammasome has been implicated in pathogenesis of many chronic liver diseases, including viral hepatitis, non-alcoholic steatohepatitis (NASH) and alcoholic liver disease [16]. NLRP3 was shown to contribute to fibrosis development in a in vivo model of NASH, and patients with increasing levels of fibrosis exhibited increased NLRP3 mRNA levels [17,18]. High levels of inflammasome components are present in many epithelial tissues, where they have been shown to represent an important first line of defense [19]. Interestingly, a number of recent studies have highlighted the importance of NRLP3-dependent mucosal immunity also in inflammatory bowel diseases (IBD), which are frequently associated with cholestatic liver diseases. Moreover, single nucleotide polymorphisms in the NLRP3 gene have been associated with the development of both ulcerative colitis (UC) and Crohn’s disease [20,21].

In the present review, we summarize current knowledge on the interrelation between the gut and the hepatobiliary system, with a particular focus on cholestatic liver diseases and inflammasome activation. Disease-induced alterations of gut-derived PAMPs and biliary response to innate immunity receptor activation, including the NLRP3 inflammasome, may, indeed, be of particular relevance for the development and progression of cholangiopathies.

## 2. Activation of the Inflammasome

The sensor component of the inflammasome system is a nucleotide-binding oligomerization domain (NOD)-like receptor (NLR) [22]. NLRs contain three domains: (i) a central nucleotide-binding and oligomerization domain (NACHT) responsible for oligomerization; (ii) a C-terminal leucine-rich repeat (LRR), capable of recognizing a specific ligand; and (iii) an effector-variable N-terminal interaction domain. The N-terminal domain can be of three types, i.e., a caspase recruitment domain (CARD), pyrin domain or baculoviral inhibition of apoptosis protein repeat domain (BIR) [22]. Depending on the effector domain, it is possible to classify the inflammasomes into three subfamilies: the NLRP, which contain a pyrin domain, the NLRB, which contains a BIR domain, and the NLRC in which a CARD is present. To date, a total of 22 members of the NLR protein family have been reported [22]. NOD1 and NOD2 are CARD-containing NLRs that lead to the activation of CARD9 and nuclear factor (NF)-kB pathways interacting with the receptor-interacting serine/threonine-protein kinase 2 (RIPK2) [23]. In contrast, several NLR proteins containing a pyrin domain were found to form a signaling platform, the well-known inflammasome, driving caspase activation by binding to the adaptor protein, the apoptosis-associated speck-like protein containing a CARD (ASC) [24,25]. Instead, the NLR apoptosis inhibitory proteins (NAIPs), containing the BIR domain, function as specific cytosolic receptors for bacterial ligands to form the NAIP–NLRC4 inflammasome for anti-bacterial defenses [26].

Among the various prototypes of inflammasomes known, the NLRP3 inflammasome is the most extensively studied (Figure 1) [27].

The NLRP3 inflammasome is typically composed of the inflammasome sensor NLRP3, the adaptor molecular apoptosis-associated speck-like protein containing a caspase-recruitment domain (ASC), and the precursor pro-caspase-1 [27,28]. When activated, the assembly of the NLRP3 inflammasome platform causes the activation of caspase-1, which leads to the maturation and secretion of proinflammatory cytokines including IL-1β and IL-18 [13,29,30]. The expression of NLRP3 itself seems to be the limiting factor for the activation of the NLRP3 inflammasome [31]. To date, a two-step model has been well established in the activation of the NLRP3 inflammasome. For the platform to be activated it requires a priming signal, which induces NLRP3 and pro-IL-1β upregulation, and a prior or coincident second signal, which is pivotal for creating a functional inflammasome [32,33]. The priming signal involves the activation of nuclear factor kappa B (NF-κB), which occurs through the stimulation of receptors such as TLR4, NOD2, TNFR, and IL-1R, normally caused by PAMPs or damage-associated molecular patterns (DAMPs). NF-κB leads to increased synthesis of pro-IL-1β and NLRP3 by binding to its promoter [31]. Furthermore, this signal involves a post-translational regulation of inflammasome components, including NLRP3 de-ubiquitination as well as SYK- and JNK-dependent ASC phosphorylation and linear ubiquitin assembly complex (LUBAC)-mediated ASC ubiquitination [34]. The second signal for NLRP3 inflammasome activation depends on various structurally dissimilar agonists, including environmental crystalline pollutants like silica, asbestos, crystalline monosodium urate or pathogen-derived ligands, like pore-forming toxins, such as nigericin [33]. This process involves various concomitant molecular mechanisms. First, an intracellular K+ efflux occurs [35], boosted by extracellular ATP that binds to P2X purinoceptor 7 resulting in large pore formation on the cell membrane mediated by the translocation of the pannexin-1 channel [36]. As a consequence of membrane permeability, PAMPs or DAMPs enter the cell and activate the NLRP3 inflammasome. Lysosomal destruction of large molecules due to the phagocytosis results in intracellular release of its components, which are also able to activate the NLRP3 inflammasome [37]. Additionally, the effect of reactive oxygen species (ROS) derived from mitochondria leads to thioredoxin dissociation, which binds to the NLRP3 inflammasome to trigger its activation [38]. 

## 3. The Gut–Liver Axis in Cholangiopathies

A strong interrelation between the hepatobiliary system and the intestine exists both in physiological and pathological conditions (Figure 2).

Among other functions, the liver is responsible for bile secretion, which reaches the gut via the biliary tree. Bile acids, which form the majority of the excretory products of bile, together with bilirubin, are essential in the digestive process of lipids and fat-soluble vitamins but are also involved in the modulation of the gut microbiota composition, especially in the small intestine [39]. Indeed, bile acids are able to directly damage bacterial cell membranes with an efficacy that mainly depends on concentration and hydrophobicity [40]. It appears intuitive that diseases that alter the physiological bile flow, such as chronic cholestatic liver diseases, may deeply influence gut bacterial composition. Small intestinal bacterial overgrowth has been demonstrated in PBC [41] and cirrhotic patients [42]. Moreover, a number of studies in animal models have shown that bile duct ligation causes bacterial overgrowth with increased bacterial translocation, which could be reversed by oral administration of bile acids or whole bile [43,44,45]. 

Bile acids are produced in hepatocytes as end products of cholesterol metabolism. The two primary bile acids, namely cholic acid and chenodeoxycholic acid, are conjugated with glycine or taurine to increase water solubility and are excreted in the bile canaliculus. Conjugated bile acids are then actively reabsorbed, predominantly in the terminal ileum, by active transport via the apical sodium-dependent bile acid transporter (ASBT), reach the liver and are secreted again in bile, in a process known as entero-hepatic circulation [39]. When the primary conjugated bile acids reach the terminal ileum and colon, the resident bacterial community extensively metabolizes them, giving rise to a wide array of secondary bile acids via deconjugation, dehydrogenation, and dehydroxylation reactions [46]. Secondary bile acids, which form the vast majority of the bile acids in the colon, may also be passively reabsorbed and enter the entero-hepatic circulation. As a result of this extensive metabolism, the final bile salt pool is deeply influenced by the microbiota composition of the individual and more than 50 different secondary bile acids are found in feces [47,48]. Moreover, the bile acid pool is involved in the feedback mechanism of bile acid synthesis. Indeed, bile acids bind with different affinity with the farnesoid X receptor (FXR), expressed in both the hepatocytes and enterocytes. FXR activation represents the cornerstone of bile acid homeostasis, downregulating bile acid synthesis, transport and reabsorption [49]. Interestingly, Inagaki et al. have recently demonstrated that bile acids also exert indirect antimicrobial effect via FXR-dependent expression of several genes involved in mucosal defense [50].

In a physiological state, the dynamic equilibrium between the microbiota composition and the bile acid pool ensures that beneficial effects are exerted on multiple levels by commensal bacteria and bile acid-related endocrine functions [51,52]. Moreover, the large amounts of PAMPs that reach the liver via the portal flow, which may activate a number of receptors of innate immunity [53,54], are kept at bay by a tightly regulated immune tolerance, depending on hepatic antigen-presenting cells, such as Kupffer cells, liver sinusoidal endothelial cells and hepatic stellate cells [55]. Cholangiocytes, too, have been shown to display immune tolerance toward LPS, mainly via the upregulation of IRAK-M (a negative regulator of TLR signaling) [56]. A disruption of the balance in the microbiota composition, responsible for the development of dysbiosis, may in turn have deleterious effects on the hepatobiliary system. 

The pathogenesis of PSC represents a clear demonstration of the close association between dysbiosis, intestinal permeability and cholangiocyte injury. In fact, previous studies have shown that about 70% of PSC patients are also affected by IBD, with a distinct form of colitis different to ulcerative colitis (UC) [57]. On the contrary, about 5% of UC patients develop PSC during the course of the disease, suggesting that intestinal inflammation and dysbiosis alone are not sufficient to cause cholangiocyte injury [58]. Colectomy before or at the time of liver transplantation (LT) significantly reduces the risk of PSC recurrence, which is reported to be as high as 37% after LT [59,60]. Antibiotic treatment has been recently confirmed to be effective in improving serological markers of cholestasis in PSC patients, with a possible indirect effect via modifications of the gut microbiome [61]. Moreover, a number of markers of bacterial translocation and gut barrier dysfunction (i.e., zonulin, intestinal fatty acid binding protein, soluble CD14, LPS and LPS-binding protein, antibodies against F-actin and gliadin, and various anti-microbial antibodies) have been found elevated in the sera of PSC patients, with a positive correlation with progressive disease [62,63]. Interestingly, genome-wide association studies have revealed a common risk factor for PSC and Crohn’s disease in the fucosyltransferase 2 (FUT2) locus, which influences fecal and bile bacterial composition and has recently been linked to the development of hepatobiliary abnormalities in mice [64,65]. 

Despite less robust, preliminary evidence of the possible role of gut-derived PAMPs has also emerged for PBC. Lipoteichoic acid, a cell wall component of gram-positive bacteria, has been detected in the portal tracts of PBC patients, with a profile of immunoreactivity that changes according to disease stage [66]. Anti-mitochondrial antibodies (AMA), the diagnostic hallmark of PBC, display crossreactivity against E. coli, which are enriched in the feces of patients compared with controls [67,68]. Moreover, PBC patients have higher levels of serum LPS and hepatic expression of TLR4, CD14, CD68 and NF-κB when compared to control individuals [69].

Interestingly,,a number of studies have reported sharp differences in the microbiota composition of PSC and PBC compared to healthy individuals, despite it is still not clear if they are cause or consequence of the biliary disease [54,70]. Typically, the microbiota is mainly composed of two phyla, Firmicutes and Bacteroides, while the Proteobacteria accounts for up 2%–3% of the gut milieu [71]. Alterations in the intestinal microbiota have been described in PSC patients, with or without concomitant IBD [72]. At the genus level, PSC patients had an increased abundance of Enterococcus, Lactobacillus, Streptococcus and Fusobacterium genera compared to healthy individuals and independent of IBD [73]. The Veillonella genus, which is also associated with other chronic inflammatory and fibrotic conditions, was enriched in PSC [74]. As in PSC, a microbial signature consisting of 12 genera has been identified for PBC [75]. In PBC patients, microbiota profile analysis highlighted the enrichment of the Enterobacteriaceae family, followed by Pseudomonas, Veillonella and Clostridium genera; in reverse, Oscillospira and Sutterella were less represented [76]. 

A number of in vivo studies in animal models also support the close relationship between the hepatobiliary system and the gut in disease development and progression. Long-term parenteral inoculation of high amounts of specific strains of bacteria has been shown to cause non-suppurative cholangitis in BALB/c mice [77]. Experimental colitis can be induced in mice by oral administration of dextran sodium sulfate (DSS), which induces mucosal injury with increased intestinal permeability [78]. A previous study demonstrated that hepatic cytochrome P450 expression is reduced in rats subjected to DSS-induced colitis via LPS-induced downregulation [79]. Patients affected by cystic fibrosis lack the function of the cystic fibrosis transmembrane conductance regulator (CFTR), which is involved in bile flow and alkalinization, and may develop liver damage due to chronic cholestasis [80]. DSS-induced colitis has been demonstrated to cause biliary injury in *Cftr* knockout mice, which could be reversed by antibiotic treatment, and not by bile induction, via administration of 24-nor-ursodeoxycholic acid [81]. In *Mdr2* knockout mice, a commonly used murine model of PSC, the development of a ductular reaction, fibrosis and ductopenia was significantly higher in germ-free (GF) *Mdr2* knockout animals compared with conventionally raised mice [82]. As expected, secondary bile acids were absent in GF *Mdr2* knockout mice, while cholangiocyte senescence was significantly increased in vitro and could be reduced by treatment with the secondary bile acid ursodeoxycholic acid [82]. The NOD.c3c4 mouse model spontaneously develops spontaneous lymphocyte infiltrations around the bile duct and AMA, closely resembling human PBC [83]. Schrumpf et al. have recently demonstrated that NOD.c3c4 mice have a distinct gut microbiota when compared to control mice and this has a clear effect on the development of the biliary phenotype. Indeed, GF NOD.c3c4 mice developed milder biliary alterations compared with conventionally raised NOD.c3c4 mice. A similar effect could be obtained also by antibiotic treatment of NOD.c3c4 mice [84]. Taken together, these in vivo experiments show that, in the context of a permissive genetic background (Cftr knockout, Mdr2 knockout, NOD.c3c4 mice), the alteration of intestinal permeability and/or gut microbiota is pivotal in inducing or maintaining biliary inflammation. In an effort to devise effective treatments, future research will necessarily be aimed at determining if biliary alterations are the cause or consequence of microbiota and intestinal permeability alterations, especially in the setting of human cholangiopathies.

## 4. Inflammasome Activation in Cholestatic Liver Injury

Despite definitive mechanistic proof that a differential activation of the NLRP3 inflammasome due to altered PAMPs delivery to the hepatobiliary system during the course of cholestatic liver disease is missing, a number of studies have started to evaluate the role of NLRP3, especially in model systems, with interesting results. In biliary atresia, a neonatal obstructive cholangiopathy, increased expression of NLRP3, Caspase-1 and IL-1R1 has been demonstrated in the livers of patients at the time of diagnosis [85]. Interestingly, in a murine model of biliary atresia, both Nlrp3^-/-^ and Il-1R1^-/-^ mice showed reduced biliary damage and inflammation compared to controls; however, Caspase-1^-/-^ mice were not protected, suggesting that inflammasome activation may result in biliary damage via the activation of non-canonical pathways in this setting [85]. Matsushita et al. showed that the expression of NLRP3 in cholangiocytes, evaluated by immunohistochemistry, is induced also in advanced-stage PSC patients when compared to those in the early-stage. Interestingly, higher levels of NLRP3 expression were also detected in PSC patients with UC than in PSC patients without UC, while patients developing cholangiocarcinoma showed lower levels of NLRP3 [86]. Increased expression of NLRP3 and Caspase-1 has been also demonstrated in the livers of PBC patients [87]. Our group has recently demonstrated that the expression of Nlrp3 and Asc is specifically induced in cholangiocytes of mice subjected to 3,5-diethoxycarbonyl-1,4-dihydrocollidine (DDC) treatment, as a model of sclerosing cholangitis [88]. In vitro, the induction of the Nlrp3 inflammasome in cultured cholangiocytes promoted the upregulation of Il-18 but not of IL-1β or Il-6 and had no effect on the proliferation of biliary cells, which is one of the primary responses to injury of cholangiocytes and is thought to be essential in disease progression and fibrosis development [89]. In contrast, activation of the inflammasome reduced the increase in epithelial permeability induced by stimulation with LPS and ATP through a regulation of E-cadherin and Zonulin-1 expression [88]. The disruption of the epithelial integrity of the cholangiocyte layer, with spillover of bile acids in the portal tract, has been demonstrated to be a key pathogenic mechanism in Mdr2^-/-^ mice, a commonly used murine model of PSC [90]. Moreover, a number of PAMPs and cytokines (LPS, TNF-α and INF-γ) have been shown to directly impair the epithelial barrier function of cholangiocytes, both in vitro and in murine models [91,92]. Interestingly, strong evidence supporting the interrelation between the gut microflora and cholangiocyte pathology has been recently demonstrated in Mdr2^-/-^ mice. Liao et al. showed that Mdr2^-/-^ mice have a particular microbiota composition that is associated with prominent activation of the Nlrp3 inflammasome. Moreover, transfer of Mdr2^-/-^ mice microbiota in wild-type mice was sufficient to cause hepatic damage, which was significantly reduced by concomitant administration of a pan-caspase inhibitor [93].

Activation of the inflammasome in immune cells may also contribute to cholestatic disease progression via complex bile acid-induced modulations of NLRP3 functions. Guo et al. recently demonstrated that the activation of NLRP3 in macrophages is inhibited by incubations with different bile acids, with the most potent effect exerted by lithocholic acid [94]. With a series of in vitro and in vivo experiments, the authors demonstrated that bile acid-induced inhibition of the inflammasome is mediated by the activation of bile acid receptor transmembrane G protein-coupled receptor-5 (TGR5) in the cell membranes of macrophages, with subsequent protein kinase A (PKA)-dependent phosphorylation and ubiquitination of NLRP3 [94]. Such a mechanism may limit inflammatory response in the setting of cholestasis. However, conflicting effects of bile acids on NLRP3 activation have also been described [95]. In the setting of cholestasis, increased levels of bile acids activated both the primary and secondary signals for NLRP3 activation via inducing a prominent calcium influx in macrophages. The authors also demonstrated that FXR acted as a negative regulator of NLRP3 activation; indeed, wild-type mice transplanted with the bone marrow of Fxr^-/-^ mice showed increased plasma levels of IL-1β after LPS treatment when compared to control-transplanted mice [95]. In support of this data, chenodeoxycholic acid has been shown to induce NLRP3 activation in macrophages and Kupffer cells, possibly through TGR5/EGFR-dependent ROS formation, and to contribute to the liver fibrosis in the bile duct ligation murine model [96]. However, because of the high concentrations of hydrophobic bile acids used in some experiments, some authors have questioned that bile acids can directly activate the inflammasome in hepatocytes or macrophages [87]. Cai et al. showed that, while Caspase-1^-/-^ mice subjected to bile duct ligation (BDL) had more severe liver fibrosis, the incubation of hepatocytes or macrophages with cholestatic levels of conjugated bile acids did not stimulate the activation of the inflammasome, suggesting an indirect effect of NLRP3 activation in liver damage [87]. 

Initial reports have also addressed possible targets for therapeutic interventions that may interfere with inflammasome activation. The intraperitoneal administration of MCC950, a specific small molecule inhibitor of NLRP3, resulted in the amelioration of liver injury, survival and fibrosis in BDL-treated mice, strongly supporting a possible therapeutic modulation of the inflammasome [97]. Previous studies have shown that Galectin-3, a lectin produced by macrophages, may activate the NLRP3 inflammasome in the liver and contribute to collagen deposition in a murine model of cholestasis [98]. Interestingly, treatment of mice with Davanat, a registered inhibitor of Galectin-3 that is currently being studied in clinical trials, determined a significant reduction in inflammasome activation and biliary damage in a model of autoimmune cholangitis [99]. Vardenafil, a phosphodiesterase-5 inhibitor, has been shown to significantly reduce the expression of Nlrp3 and inflammasome components in a cholestatic murine model induced by lithocolic acid administration [100]. Finally, a significant reduction in Nlrp3 inflammasome activation has been demonstrated in the liver tissue of BDL rats subjected to treatment with a saline solution enriched with methane, which has been previously demonstrated to possess anti-oxidative and anti-inflammatory properties [101,102].

## 5. Conclusions

The gut and the liver are deeply connected in both physiological and pathological conditions. With respect to cholangiopathies, this is clearly exemplified not only in a number of in vivo models of biliary injury, but also in PSC patients, who are frequently affected by IBD. Despite the fact that the cause or consequences of the disease are currently unclear, microbiota alterations specific for PBC and PSC have been demonstrated in a number of studies and may affect the development of the disease [70,76]. A variety of bacterial products and PAMPs, in the context of disease-specific dysbiosis, have, indeed, been shown to exert deep effects on the physiology of cholangiocytes through the activation of TLRs and, more recently, activation of the NLRP3 inflammasome. A number of in vitro and in vivo experiments have started to shed light on the molecular mechanisms behind NLRP3 activation and regulation in the context of cholestatic liver disease. Future experiments need to specifically evaluate if disease-induced dysbiosis exerts a direct effect on NLRP3 activation in the hepatobiliary system and, if possible, define the specific molecular triggers in this context. A deeper understanding of such mechanisms and their fine-tuning is expected to offer new therapeutic approaches for patients, especially in the era of precision medicine.

## Figures and Tables

**Figure 1 cells-09-00736-f001:**
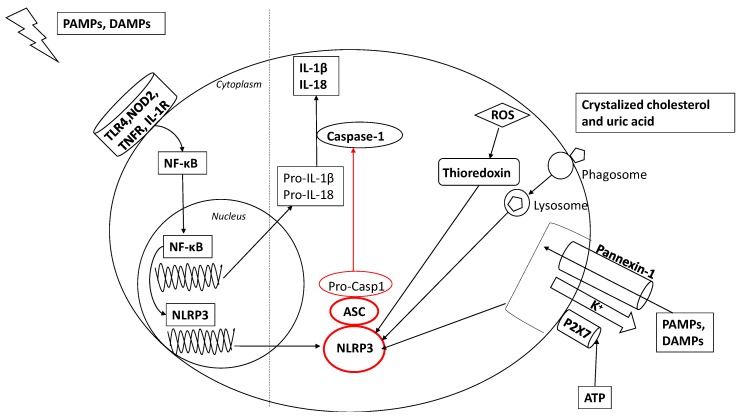
Two-step model of nucleotide-binding oligomerization domain (NOD)-like receptor pyrin domain-containing-3 (NLRP3) inflammasome activation. In the first step, pathogen-associated molecular patterns (PAMPs) and damage-associated molecular patterns (DAMPs) activate the nuclear factor (NF)-κB pathway through the stimulation of receptors such as toll-like receptor (TLR)4, nucleotide-binding oligomerization domain-like receptor (NOD)2, tumor necrosis factor receptor (TNFR), and interleukin (IL)-1R. In addition, NF-κB activates NLRP3 gene transcription. The second step involves various concomitant molecular mechanisms. Extracellular ATP induces P2×7-dependent pore formation on the cell membrane, which allows K+ efflux depletion and opening a pannexin-1 channel, through which PAMPs and DAMPs enter in the cell, activating NLRP3. Endocitosis of dissimilar agonists, including crystalized cholesterol and uric acid, results in lysosomal disruption, which also leads NLRP3 activation. Additionally, reactive oxygen species (ROS) leads to thioredoxin dissociation, which binds to the NLRP3 inflammasome to trigger its activation. The NLRP3 inflammasome is composed of an inflammasome sensor NLRP3, apoptosis-associated speck-like protein containing a CARD (ASC) and the precursor pro-caspase1: when activated, NLRP3 components cause the activation of caspase-1, which leads to the maturation and secretion of proinflammatory cytokines including IL-1β and IL-18.

**Figure 2 cells-09-00736-f002:**
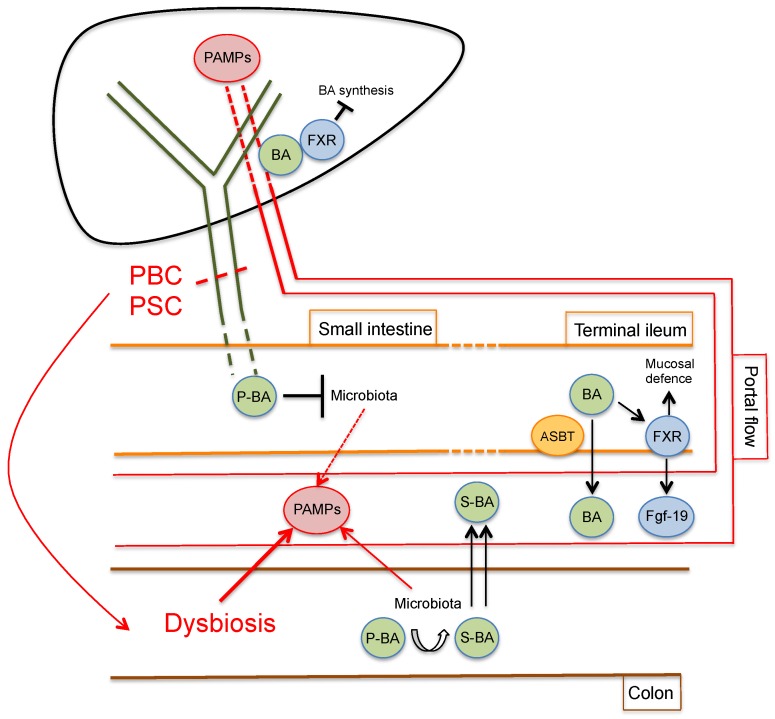
Gut–liver axis during cholangiopathies. In normal conditions, primary bile acids (P-BA) reach the small intestine where they directly influence the microbiota composition, mainly by blocking bacterial overgrowth. BA are actively reabsorbed in the terminal ileum via the apical sodium-dependent bile acid transporter (ASBT) and activate farnesoid X receptor (FXR), which, in turn, causes Fgf-19 secretion and direct effects on mucosal defense. In the colon, bacterial metabolization produces a wide variety of secondary bile acids (S-BA), which also enter the portal circulation. Cholangiopathies, which alter the normal bile flow or composition, may interfere with all these processes, ultimately causing dysbiosis and an increased or qualitatively altered PAMPs delivery to the liver via the portal circulation.

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
