# Peer review of "Gut–Liver Axis and Inflammasome Activation in Cholangiocyte Pathophysiology"

_cells, 2020, doi:10.3390/cells9030736_

Round 1

Reviewer 1 Report

The submitted review explains in good way the role of NLRP3 in gut-liver axis in case of inflammation, and how changing in the gut permeability and changing in microbiota composition could modify an important equilibrium and have a role in biliary and liver disease.

In my opinion, to enrich the submitted review authors should improve some points:

  • Authors should describe more in detail the three terminal N domains of NLR, giving at least one example for each typology.
  • Authors should indicate the most important microbiota components that have a greater change in concentration due to biliary/intestinal pathologies.
  • Author could describe if in the literature are present studies on intestinal pathologies (for example Chron's disease or ulcerative colitis) that have analysed the expression of NLRP3 in cholangiocytes.

Moreover, authors should correct some mistakes found into manuscript:

  • Page 3, line 101, authors should change posttranslational with post-translational.
  • Page 5, line 166, authors opened a bracket but after they did not close.
  • Page 5, line 199, authors have not described the full name of CFTR.
  • Page 7, line 270, authors have not described the full name of BDL.

Finally, I would like to suggest to put in bold the title in figure legends in order to make easier the read of these.

Author Response

We thank the Reviewer for the attention given to our paper in order to improve the quality of the manuscript.

We have addressed all the comments and the issues raised by the Reviewer.

- In response to the first comment of the Reviewer, we have added a brief section about the terminal domains of the NLR. This part can be found at lines 79 – 86.

- As requested from the Reviewer, we have added the most important studies regarding the differences in the microbiota composition in PSC and PBC. We have also provided references to recent reviews dealing in details with this subject. This part has been added from line 208 to line 219.

- As the Reviewer correctly suggests, there are a number of published papers dealing with the role of NLRP3 in pathogenesis of IBD. Apart from NLRP3-dependent secretion of a number of cytokines involved in the regulation of mucosal immune response, single nucleotide polymorphisms in the NLRP3 gene have been associated with ulcerative colitis and Crohn’s disease. However, to the best of our knowledge, none of them investigated specifically the expression of NLRP3 in cholangiocytes of patients affected by IBD or in murine models of IBD. We have therefore added in the manuscript a brief sentence, indicating that NRLP3 is involved in the pathogenesis of IBD and suggested recent literature on the topic for the reader (line 60 – 64).

Finally, we have also modified the text in accordance to the reviewer suggestion for mistakes and typos.

We did not modify the figure legend layout since this was prepared in accordance with the template provided by the Editor; we are, however, ready to modify it if necessary.

Reviewer 2 Report

Dear authors,

thank you for letting me participate in the review of your manuscript cells-733904 'Gut-liver axis and inflammasome activation in cholangiocyte pathophysiology. I don't have additional comments other than those minor thoughts documented in attached pdf file. 

Author Response

We thank the Reviewer for the attention given to our manuscript.

We have corrected all the mistakes pointed out by the Reviewer in the revised version of the manuscript.

Following the suggestion of the Reviewer, we have also corrected the inaccuracy at line 186 of the revised version of the manuscript.

Reviewer 3 Report

In this review article, the authors summarized current knowledge regarding roles of gut-liver axis and the inflammasome activiation in the setting of cholangiopathies.  Overall, this review is well prepared, and included basic knowledge of the inflammasome activation, roles of gut-liver axis in cholangiopathies, and roles of the inflammasome activation in cholestatic liver injury.  This review potentially can be interesting and useful to researchers who are working in the area of cholangiopahties. 

The major drawback of this manuscript is lack of direct evidence which shows a connection between gut microbiota and activation of the hepatic inflammasome.  This drawback makes different parts of this review are not logically related.  The authors need to add anther section to discuss the connection between gut and hepatic inflammasome activation, and also explain the reasons why put gut-liver axis and the activation of inflammasome together in this review.

Author Response

We thank the Reviewer for the attention given to our manuscript and for recognizing the potential interest of the readers for our review.

In response to the Reviewer comment, we acknowledge that currently there is no direct evidence linking the activation of the NLRP3 inflammasome and the alterations of the gut microbiota in the specific setting of cholangiopathies. In particular, as the Reviewer correctly suggests, to the best of our knowledge there is no direct proof that, during the course of chronic cholestatic diseases, dysbiosis-induced alterations of the wide array of bacterial products that reach the liver (and the cholangiocytes) determine a differential activation of the NLRP3 inflammasome as compared to control or healthy conditions. We have specified this aspect in the manuscript and modulated our claims (lines 249 – 252 and 328 – 332).

Bearing the aforementioned notions in mind, we have tried to summarize available literature on the topic and proposed a concise overview of the NLRP3 molecular activation mechanisms, followed by the interrelations between the hepatobiliary system and the gut microbiota (that may modify the composition of NLRP3-activating bacterial products reaching the liver, especially during disease-induced dysbiosis) and finally reported available data on inflammasome activation in cholestatic liver injury. We acknowledge that, in some instances, the review may miss conceptual links and additional evidence to fully support a possible role of the NLRP3 inflammasome in cholestatic diseases. Hopefully, future studies will help improving current flaws.

Reviewer 4 Report

This review by Maroni et al. is dealing with gut-liver axis and inflammasome activation in cholangiopathies.

It is very well written and concise and of clinical and scientific relevance!

The authors may ad the paper by Wree A et al. Metabolism 2014 which is describing the role of adipose tissue and NAFLD, which support authors view!

Author Response

We thank the Reviewer for the attention given to our manuscript and for acknowledging the quality of our work.

In accordance with the Reviewer suggestion, we have added a brief mention regarding the role of NLRP3 in NASH and added the work of Wree et al. as suggested. This section can be found from line 54 to line 58 of the revised version of the manuscript.

Round 2

Reviewer 3 Report

The manuscript has been significantly improved.  I agree it for publication in present form.